# Lifecycle Assessment for Recycling Processes of Monolayer and Multilayer Films: A Comparison

**DOI:** 10.3390/polym14173620

**Published:** 2022-09-01

**Authors:** Gerald Koinig, Elias Grath, Chiara Barretta, Karl Friedrich, Daniel Vollprecht, Gernot Oreski

**Affiliations:** 1Chair of Waste Processing Technology and Waste Management, Department of Environmental and Energy Process Engineering, Montanuniversitaet Leoben, Franz Josef-Straße 18, 8700 Leoben, Austria; 2Polymer Competence Center Leoben GmbH, Roseggerstraße 12, 8700 Leoben, Austria; 3Chair of Resource and Chemical Engineering, Augsburg University, Am Technologiezentrum 8, 86159 Augsburg, Germany

**Keywords:** 2D plastic packaging, near-infrared spectroscopy, sensor-based sorting, life cycle assessment, small film packaging

## Abstract

This work covers a lifecycle assessment of monolayer and multilayer films to quantify the environmental impacts of changing the management of plastic film waste. This lifecycle assessment offers the possibility of quantifying the environmental impacts of processes along the lifecycle of monolayer and multilayer films and mapping deviating impacts due to changed process parameters. Based on the status quo, the changes in global warming potential and abiotic fossil resource depletion were calculated in different scenarios. The changes included collecting, sorting, and recycling mono- and multilayer films. The “Functional Unit” under consideration comprised 1000 kg of plastic film waste, generated as post-consumer waste in Austria and captured in the lightweight packaging collection system. The results showed the reduction of environmental impacts over product lifecycles by improving waste management and creating a circular economy. Recycling all plastic film reduced global warming potential by 90% and abiotic fossil resource consumption by 93%. The necessary optimisation steps to meet the politically required recycling rates by 2025 and 2030 could be estimated, and the caused environmental impacts are presented. This work shows the need for increased collection, recycling, and significant improvement in the sorting of films to minimise global warming potential and resource consumption.

## 1. Introduction

Plastics are omnipresent in everyday life, and their primary areas of application can be found in the packaging industry, the construction industry and the automotive industry [1]. Plastics as packaging material reduces packaging mass, energy consumption and greenhouse gas emissions [2]. In the packaging industry, in particular, plastic products are required, which, in addition to the requirements for the protection of the packaged products and industrial processability, must also meet the optics, haptics and consumer information.

The legal framework in the European Union and Austria creates the basis for an increased focus on the recycling of plastic packaging. The legal framework for plastics and packaging in the European Union and Austria, defined by laws, regulations, directives, strategies and action plans, sees regulations such as the EU circular economy package, the EU plastics strategy, the EU single-use plastics directive, the EU packaging directive 94/62/E.G. and the Austrian Waste Management Act 2002 propose increases concerning plastics recycling.

Along with the increasing annual volume of plastic waste of around 19% in the period from 2006 to 2018 [1] and the policy instrument to realise the potential of waste for the provision of secondary raw materials through mandatory recycling quotas [3], optimisations in sorting and recycling processes are necessary.

Increasing these recycling quotas is an ecologically viable way to fulfil ecological requirements by increasing the recycling of currently ignored materials streams such as multilayer films, which make up a substantial portion of lightweight packaging in Austria and are currently being incinerated, which adds to the carbon footprint and squanders valuable resources. Considering that of 300,000 t of plastic packaging waste generated per year in Austria small films account for 69,000 t, the potential for improvement in this area is substantial [4,5]. These films are commonly made from polyolefins, like polyethylene (PE), polypropylene (PP), or other polymers such as polyethylene terephthalate (PET). Further in the multilayer materials, various combinations of polymers, such as polyamide and PE or PE and PP are common.

Despite the current difficulty in recycling mono- and multilayer packaging, techno-economic analysis of processes to separate multilayer materials and use the separated polymers for recycling show promise. Considering that recycled polymers must be produced at a price similar or lower than virgin materials to be economically viable and simultaneously have to require less resources such as energy in their production to be considered ecologically sensible, the creation of processes that satisfy all these requirements is currently undergoing. Amongst these are APK’s Newcycling [6], Unilever’s CreaSolv [7] and the STRAP [8] process. The STRAP process showed a 37% reduction in energy requirement for separating the layers of a multilayer, recovering the PET contained therein, compared to the energy required in the production of virgin PET [8].

Recycling monolayer films is equally difficult because foreign materials may be introduced to an otherwise-clean mono-material waste stream. This contamination can occur by introducing multilayer films into the feedstock, which contain foreign materials and thus can have a detrimental effect on recyclability [9]. To remove these contaminants or subsequently receive them as a separate recyclable fraction, optimisations in the waste management processes of plastic films are necessary.

Instruments for quantifying the changes in environmental impact are required to justify necessary optimisations and gauge their possible impact. In particular, the lifecycle assessment (LCA) is applicable. An LCA quantifies and represents the lifecycle or changes in processes throughout the product’s lifecycle [10].

Existing LCAs include a study conducted by Choi et al. from 2018, who investigated the carbon footprint of packaging films made from low-density polyethylene (LDPE), polylactic acid (PLA), and PLA/polybutylene adipate terephthalate blends (PLA/PBAT) in South Korea. The results of this LCA show that incineration, as the waste treatment measure with the highest global warming potential (GWP), has the worst balance sheet for each plastic fraction considered [11].

Volk et al. carried out an additional LCA in 2021, representing a techno-economic assessment and comparison of different plastic recycling pathways. The study done by Volk et al. (2021) evaluates the effects of different recycling routes of separately collected lightweight packaging in Germany. The GWP, the cumulative energy requirement (CED), the carbon efficiency and the product costs were considered by Volk et al. The recycling routes included mechanical recycling, chemical recycling, and a combination of both methods. The study shows that incineration accounts for 95% of the GWP impact of the recovery route. By combining mechanical and chemical recycling, around 0.48 kg CO_2_-eq per kg input of waste can be saved concerning the current situation of the recycling route in Germany [12].

It remains to determine whether replacing the incineration and thus the generation of energy by recycling processes and subsequent reduction of virgin materials benefits the environment. An increase in transportation and collection efforts to manage the increase in separately collected lightweight film packaging and the increased effort to recycle these materials may offset the potential benefits of reduced thermal recovery.

Further, technological limitations may hinder the implementation of a circular economy of films packaging, as the amount of recyclate which can be introduced to new products is limited.

The LCA conducted in the here-presented work considers the effect improved collection, sorting and recycling of lightweight two-dimensional packaging can have on the GWP and abiotic resource depletion fossil (ADPF) in the Austrian waste management sector. To this end, scenarios reflecting different collection and recycling models of monolayer and multilayer materials are depicted and compared to the status quo. These scenarios entail separating monolayer materials and using them as a value-adding feedstock for mechanical recycling, thus reducing the need for thermal recovery and virgin material.

These scenarios show latent potential for saving resources and greenhouse gas emissions through improved sorting and increased recycling of mono- and multilayer films. The ADPF has been calculated for each scenario. The ADPF mirrors the consumption of fossil fuels such as oil or gas and subsequent depletion of non-renewable abiotic resources. The consumption of these resources represents the environmental impact of the production of virgin packaging materials.

To conclude, it may be stated that the sensibility of replacing thermal recovery with other means of use depends on the comparison of the environmental impact of each after-use recovery method. This article aims at aiding in the building of a fundamental basis for discussion for the implementation of future policies and the prioritisation in the development of recovery techniques by showing whether a reduction of GWP and ADPF is indeed feasible with the recycling of packaging film and by providing an estimate as to the possible reduction in these metrics with different measures taken.

## 2. Materials and Methods

In the following, the used materials for conducting the LCA and the used software are explained.

### 2.1. Current Status Survey of the Situation Regarding the Occurrence and Treatment of Plastic Waste in Austria

Prior to the LCA, comprehensive research of the current situation regarding the generation and treatment of plastic waste in Austria was conducted. The collected data form the basis for the LCA. Further, the collected data enable a comparison between the current status with the alternative scenarios, namely the GWP and the depletion and use of non-renewable and renewable abiotic resources or fossil abiotic resource depletion, in short ADPF. Both gauges were calculated in the course of the LCA. Data from Statistics Austria, Eurostat and existing literature representing the current status were collected during this preliminary work.

### 2.2. Software Used

#### 2.2.1. subSTance flow Analysis

The freeware subSTance flow Analysis (STAN), Version 2.6.801 by the Research Unit of Waste and Resource Management at TU Wien (Vienna, Austria) was used for the computation of the LCA. STAN offers the user a platform for presenting material flow analyses according to ÖNORM S2096. STAN allows the user, after complete modelling of the material flows, to automatically calculate unknown variables.

#### 2.2.2. GaBi (Holistic Accounting) (Education Licence)–Version 9.2.1.68

The LCA software GaBi was used for the LCA presented in this manuscript. The software enables the modelling of complex processes with automatic tracking of material and energy and emission flows, and its implemented databases allow access to data for modelling. The used databases are mentioned throughout the manuscript. GaBi allows for assessment methods for quantifying environmental impacts. The software also supports the user in displaying results. It has been used to create flow diagrams for each scenario.

### 2.3. Conduction of the Lifecycle Assessment

The presented LCA examines the environmental impact caused by plastic films throughout their lifecycle. The focus of this study was the analysis of the environmental impact the improved separation and subsequent recycling of mono- and multilayer films has. This improved separation is enabled by adapting current sorting methods to enable classification of polymer films and allows for a material utilisation of foil materials which are currently primarily recovered thermally.

#### 2.3.1. Functional Unit

The considered “functional unit” comprised 1000 kg of plastic film waste generated as post-consumer waste in Austria, recorded in the collection and recycling system of the light packaging collection.

#### 2.3.2. Calculations and Definitions

The LCA in this work will look at the performance of sorting plants and recycling plants and their impact on the overall metrics gauging the depletion of fossil fuels and emission of greenhouse gases. The output of these plants will be depicted by the relative mass yield of the sorting and recycling plants.

The mass yield of the sorting plant is depicted as the sorting depth, and it is calculated using the input mass and the mass of the valuable output. The respective mass yield of the beneficiation plant is calculated as the mass yield of the beneficiation plant, in short, the recycling yield.

The process of recycling the functional unit is represented using the collection rate, the sorting depth, the recycling yield and the overall recycling rate. The collection rate represents the proportion of packaging put into circulation and collected after use. This collected fraction is then transported to the sorting plant.

The success of this sorting plant is depicted in the mass yield of the sorting plant, or sorting depth in short. After sorting, the waste fractions are processed in a recycling plant. The amount of waste successfully recycled in this step is depicted as the mass yield of the beneficiation or recycling plant, in short, the recycling yield. The overall success of the packaging recycling is calculated and depicted in the recycling rate.

The following paragraphs explain these gauges and their calculation in greater detail.


Definition of Collection Rate:


The collection rate has been calculated using the quotient of thin-layered plastic packaging produced and collected as shown in Formula (1). The collection rate represents the effectiveness of the waste collection scheme and its effectiveness in collecting thin-layered plastic film packaging. Improving the sorting discipline of the consumer and reducing sinks such as littering raises the collection rate and facilitates subsequent processes such as sorting and recycling.

In accordance with the Austrian packaging ordinance 2014, packaging put into circulation refers to the amount of packaging handed over to the end consumer (Packaging sold by the final distributor).
(1)$$Collection\;Rate\;\left[\%\right]=\frac{Collected\;Post\;Consumer\;Packaging\;\left[\frac{t}{a}\right]}{Packaging\;put\;into\;circulation\;\left[\frac{t}{a}\right]}\times 100\;$$


Definition of Sorting Depth:


The mass yield of the sorting plant, henceforth referred to as sorting depth in this work, represents the success rate when sorting the functional unit of thin-layered plastic films into the categories monolayer (S-MO) and multilayer (S-MU). This mass yield of the sorting plant is calculated for each of the two fractions generated at the sorting plant [13]. The sorting depth has been calculated as shown in Formula (2).
(2)$$Mass\;yield\;of\;sorting\;plant\;\left[\%\right]=\frac{Sorted\;fraction\;\left[t\right]-Output\;of\;sorting\;plant\;\left[t\right]}{Input\;of\;sorting\;plant\;\left[t\right]}\times 100\;$$


Definition of Recycling Yield:


The recycling yield is the quantitative proportion of a target product obtained concerning the total input flow of a recycling plant. This yield is the mass yield of valuables from the recycling plant. Improving the recycling process increases the number of valuable resources recovered from the input stream and facilitates the substitution of virgin materials with recycled polymers. The calculation is shown in Formula (3).
(3)$$Recycling\;Yield\;\left[\%\right]=\frac{Recycled\;Fraction\;\left[t\right]-Output\;of\;recycling\;plant\;\left[t\right]}{Input\;of\;recycling\;plant\;\left[t\right]}\times 100\;$$


Definition of Recycling Rate:


According to Article 11a of directive (EU) 2018/851 of the European Parliament, which defines the new calculation method for the recycling rate, the recycling rate is calculated from the quotient of generated and recycled packaging waste weights, as shown in Formula (4). The amount of packaging waste produced is equated with the amount of packaging placed on the market in the same year. Packaging waste that underwent the necessary screening, sorting and conditioning processes to remove non-recyclable waste materials and was then sent to a recycling plant to be processed is represented in the formula as recycled post-consumer packaging.
(4)$$Recycling\;Rate\;\left[\%\right]=\frac{Recycelt\;Post\;Consumer\;Packaging\;\left[\frac{t}{a}\right]}{Packaging\;put\;into\;circulation\;\left[\frac{t}{a}\right]}\times 100\;$$

Figure 1 lays out the inputs and outputs of the different stages evaluated. Here, the flow of packaging waste is depicted to enhance the formulae stated above.

#### 2.3.3. The Geographical Scope of the Investigation

The geographical scope of the investigation included the cycle of plastic products in Austria, which were manufactured, processed, disposed of and returned to the cycle in Austria.

Figure 2 shows the balance area of the product lifecycle of plastic films and delimited by the system boundary. The energy supply, supply of operating resources and the transport processes involved, which were included in the balance, were omitted for clarity. The technical standard of the processes is assumed to be an average technology mix. The product lifecycle balanced using **GaBi** is shown in Appendix A.

#### 2.3.4. Scenarios

The evaluated scenarios include the status quo, improved collection, improved separation and improved recycling. In addition, scenarios depicting the stipulated recycling quotas were calculated. Further, one scenario shows the reduction in GWP and ADPF if the currently possible maximum amount of recycled granulates were used in the production of new films, reducing the need for virgin granulates as far as possible with the current state of the art. The respective representations show the changes in the material flows due to the adjustment of the collection rate, sorting depth and recycling yield. For better illustration, only the changed material flows and waste management processes are shown. The complete material flows of the scenarios can be seen in Appendix B.

##### Scenario 1: Status Quo (SQ)

In Scenario 1, the current status of the recycling of plastic films was considered. In this scenario, the goal was to separate monolayer films from multilayer films. Therefore, the monolayer fraction was considered the target fraction for the calculation while the multilayer materials were considered contaminants. Figure 3 shows the mass flow of these plastic films concerning the respective recycling processes. The composition of the functional unit and its utilisation follow the findings from van Eygen et al. in 2018 [5].

##### Scenario 2: Improved Collection (IC)

This scenario considered the complete collection of plastic film waste. Scenario 2 was compared to the status quo. As shown in Figure 4, all plastic film waste was brought into the sorting process. The sorting depth and the recycling yield were taken from the status quo (SQ) scenario. As a result, the amounts of waste from the output flows of the sorting and recycling processes to waste incineration were increased. The increase in the amount of waste collected also increased the amount of polyethene regranulate from the recycling process, which could be brought back into production, reducing the need for virgin granulates.

##### Scenario 3: Improved Sorting (IS)

This scenario presents the changed environmental impact caused by the collection of all plastic films and a maximally optimised sorting of the monolayer films. In this scenario, multilayer films were separated from the film stream and subjected to thermal recovery while the monolayer films were sorted, recycled and subsequently used as substitute for virgin material in the production of new foils. Figure 5 shows the mass flows of Scenario 3.

##### Scenario 4: Closed Material Cycle (CMC)

This scenario expanded Scenario 3 by including the multilayer film fraction as a targeted recyclable material in the sorting and recycling processes. This scenario presumes leaps in the available technology in all areas of the recycling chain. Here, the optimum of collection, separation, recycling and substitution of virgin material has been implemented. Thus, the thermal recovery has be replaced by recycling, as shown in Figure 6. The collection rate, sorting depth, and recycling yield selected in Scenario 4 represent the optimum theoretical improvement possibilities of the waste recycling processes.

##### Scenario 5: Recycling Goals 2025—Optimisation of Collection, Sorting and Mechanical Recycling (2025)

In the 2025 scenario, the necessary optimisation steps to meet the recycling rate of 50% stipulated by the EU packaging directive through the recycling of films were examined, and the resulting environmental impact was considered [3]. To reach these recycling goals, an improvement in collection, sorting and recycling was deemed necessary. These improvements formed the basis for the calculation of this scenario. The respective percentages of the recycling chain have been improved to reach the recycling goal. Furthermore, the multilayer film fraction was considered a targeted recyclable material fraction in the recycling processes. The output flow of the recycling process thus corresponded to a total quantity of regranulate of 500 kg. Figure 7 shows the changed mass flows of the 2025 scenario.

##### Scenario 6: Recycling Goals 2030—Optimisation of Collection, Sorting and Mechanical Recycling (2030)

The 2030 scenario included determining the changing environmental impact based on the necessary optimisation steps to meet the recycling rate of 55% set by the EU packaging directive. The optimisation steps included increases in the collection rate, sorting depth and recycling yield. As a result, the amount of regranulate, consisting of polyethylene (PE) and polypropylene (PP), from the recycling process could be increased to 550 kg. This result is shown in Figure 8.

##### Scenario 7: Currently Possible Maximum Recycling Quantities with the Current State of the Art—State of the Art (SOA)

Jönkkäri et al. have shown in laboratory studies conducted in 2019 that the recyclates created from an input of 100% used films can exhibit mechanical properties congruent with the requirements for production [14]. These trials were conducted in a controlled environment and faced significant challenges during pretreatment and compounding [14]. Current recycling processes on an industrial scale require virgin material input alongside recycled polymers. This virgin input is used to ensure the mechanical properties of the final product and processability. These required mechanical properties vary depending on the production method and the desired use of the manufactured polymer.

To examine a scenario that represents the maximum amount of recycled material currently used in production without jeopardising the mechanical properties of the manufactured product, existing polymer films using recycled raw materials have been evaluated. This evaluation showed recycling contents ranging from 30% to 50% in recycled products. LDPE foils used in mechanically demanding applications, such as packaging film or stretch films, commonly comprise 30% recycled material. As stretch film made from LDPE represents the most common type of polymer packaging, this percentage has been chosen for this scenario [15]. Figure 9 shows the material flows between the collection, sorting and recycling stages.

#### 2.3.5. Comparative Overview of All Scenarios

In Table 1, an overview of the calculated scenarios is given. The collection rate, sorting depth and recycling yield of the respective scenario served as the basis for the changing process flows, shown in Figure 2, Figure 3, Figure 4, Figure 5, Figure 6, Figure 7, Figure 8 and Figure 9.

### 2.4. Lifecycle Inventory

In the following section, concrete data and assumptions for the LCA selected, based on the target and investigation framework conditions, are shown. Furthermore, the processes presented in the balance area are explained, and the lifecycle data assigned. Existing data concerning the lifecycle of the lightweight packaging fraction was used for the calculations and assumptions [5]. For non-existent data regarding the material linear low-density polyethene (LLDPE), data for the material LDPE were used.

#### 2.4.1. Raw Material Production

The processes of the primary material production of fossil PE and PP granules were modelled from the datasets “EU-28: Polyethylene Linear Low-Density Granulate (LLDPE/PE-LLD)” and “DE: Polypropylene granulate (PP) mix” of the balancing software GaBi. The transfer of the output flows of the processes to packaging production were modelled via a transport step. It was assumed that the entire amount of recyclate is used in the production in each scenario, and any gap between the provided amount of recyclate and the demand for input material in the production of new polymer films is filled with virgin material.

Scenario 4 considers a closed material loop. It has to be mentioned that this approach is currently unfeasible. As stated, current recycling processes need the implementation of virgin material to ensure the product’s mechanical properties. The assumption of a closed material loop was taken to show the latent potential in the recycling of films regarding the reduction of GWP and ADPF and to emphasise the need for technological innovation.

In scenario 7, the current limits in producing polymer films from a feedstock partly consisting of recyclates are addressed. In this scenario, the current limitations of state-of-the-art production of polymer films govern the proportion of recyclates in the input and thus the percentage of waste which can enter the recycling process chain.

#### 2.4.2. Packaging Producer

The packaging production was created as a unit process, including the data related to a specific process and lifecycle inventory (LCI) data. The fossil granulate flows from the production of raw materials, and the regranulate flows from the recycling process were recorded as input flows. The output flow included the produced plastic film fraction. Austrian waste flow consists of monolayer films mainly made of low-density PE or PP and multilayer films mainly made of combinations of PE/PP, PA/PP, PET/PA and polydimethylsiloxane (PDMS) [4]. The assumption of the material composition of mono- and multilayer films followed these findings, supported by the current materials used in the packaging sector. The composition of the material flow is shown in Table 2.

The calculations determined the energy requirement for producing plastic films from granules. For this purpose, the data of the plant Walter Kunststoffe GmbH–Gunskirchen were used. The input consumption was calculated concerning the output as shown in Appendix C. The plant produces plastic films on large rolls from PE granules (regranulates), and 353 kWh/t OUTPUT electricity consumption was determined.

The 353 kWh were determined from the output masses and the resource consumption of a comparable, state-of-the-art plant.

The substitution of primary raw materials with recyclates can only be realised to a limited extent due to the material requirements for the manufactured products and the quality of the recyclates [13]. In the LCA, the possibility of a complete substitution of the fossil granules with sufficient regranulate input was assumed to represent the optimum, and any production waste was not considered. Additionally, the currently possible highest recycling quota has been researched and implemented in an additional scenario to assess the possible savings in GWP and ADPF attainable with current technology.

#### 2.4.3. Trade and Consumer

The “Trade and consumer” process was added to complete the product life cycle of plastic films. It caused no substantial environmental impact or energy consumption.

#### 2.4.4. Usage Phase

The use phase’s modelling marked the product’s transition to the packaging waste and caused no substantial environmental impact and energy consumption. The collection rate described the proportion of separately collected waste concerning the total waste collected. Around 69,000 t of plastic film waste with a surface area of under 1.5 m^2^ (i.e., “small films”) are collected in Austria per year [5]. Of this 69,000 t, around 52,000 t were collected separately. The ratio of these waste quantities was assumed as the collection rate in the SQ scenario. The coverage rates of scenarios 2, 3 and 4 were chosen to consider maximum coverage optimisation. In the 2025 or 2030 scenario, the necessary increase in the collection rate to achieve the recycling rates required by the EU packaging directive of 50% and 55%, respectively, was calculated and set accordingly. In Table 3, the modelled collection rates of the respective scenarios are listed.

#### 2.4.5. Collection/Shipment–Lightweight Packaging and Municipal Waste

The collection processes were modelled as unit processes to link the use phase, sorting, and incineration. The processes did not cause a difference in environmental pollution or energy consumption between the different scenarios evaluated.

#### 2.4.6. Sorting

The sorting process separated the incoming waste stream into the desired target fractions, namely mono- and multilayer films. The sorting depth of scenario SQ was determined using system data from sorting systems from the report by Neubauer et al. (2020). The residual fraction output masses were divided by the input masses of the sorting plants and converted into the sorting depth of the target fraction, see Appendix D. This calculation has been performed for both target fractions, namely the monolayer films and the multilayer films.

From data collected by van Eygen in 2018, as shown in Appendix E. This parameter could be included in the calculation by dividing the sorting output (17,391 t) and the sorting input (51,964 t) [5]. Data for the energy consumption of the film sorting have been taken from existing plants using standard technology such as NIR sorting to attain the target fraction. These plants can be adapted to sort films by implementing measurements in transflection to improve the spectral quality of films [4]. The sorting depth of scenario SQ was adopted unchanged in Scenario 1. In Scenario 2, the effects of a supposed 100% successful sorting of the monolayer films and simultaneous rejection of the multilayer films as a sorting residue for energy recovery were considered. An ideal sorting process was assumed for Scenario 4 to ensure the maximum impact of this ideal scenario to show the latent potential in the recycling of plastic films. The 2025 and 2030 scenarios included the necessary optimisation of the sorting depth, considering improvements in the collection rate and recycling yield. The modelled sorting depths of the target fractions are listed in Table 4.

The increased mass of 2D waste processed is associated with additional emissions. These figures were determined by examining existing plants to determine their emission of CO_2_ and their electricity consumption when processing a functional unit of lightweight packaging. These numbers were then included in the LCA and shown in Appendix D.

The necessary equipment for sorting the 2D fraction in the relevant plants would be similar to the existing aggregates. The necessary implementation of additional reflectors to enable measurement in transflection to improve the spectral quality of thin polymer films to a point where sorting with existing NIR sensors is possible, as stated by Koinig et al. in 2022, was not included in the calculation because they were deemed negligible to the overall consumption [4].

The necessary sorting resources can be determined based on the report from Neubauer et al. (2020), which mentions the operational data concerning consumption and input masses of sorting plants [16]. In addition, data from the literature were considered in calculating electricity and gas consumption. The results of the calculation from Appendix D are listed in Table 5.

The resource consumption of the “Saubermacher Dienstleistungs AG” sorting system showed a value of around 46,981 kWh/t_INPUT_ after calculating the input-related electricity consumption. This power consumption represented a 697-fold increase in consumption compared to the resource consumption of the other sorting systems. A comparison of the literature values of the report from Neubauer et al. (2020) revealed a dot-comma error as the source of the error. After considering this source of error, the power consumption could be determined with a value of 46.98 kWh/t_INPUT_. This value was integrated into calculating the average power consumption.

#### 2.4.7. Recycling

The recycling process uses the pre-sorted film fractions of the desired PE and PP regranulates. Scenarios SQ and Scenario 1 viewed the pre-sorted waste flow as a monolayer film fraction contaminated by multilayer films, and subsequently the multilayer films were separated in the separation process to create a clean monolayer fraction for the recycling process. The recycling yield of the total fraction (monolayer films including multilayer films) in scenario SQ was calculated from the data in the report from Neubauer et al. (2020) calculated in Appendix F and adopted for Scenario 1 [16]. In total a recycling yield of 73% was achieved for the complete film fraction with a recycling yield of 96% for the monolayer fraction.

For Scenario 2, a recycling yield of 100% for monolayer films in the recycling plants following the ejection of the multilayer fraction in the sorting process was presumed. No material utilisation of the multilayer fraction has been implemented in this scenario. A recycling yield of 100% for mono- and multilayer films was selected for Scenario 3 to depict the closed cycle of plastic films.

In the 2025 scenario, the recycling yield was raised by 20% from SQ. This raise is necessary to achieve the recycling rate of 50% stipulated by the EU packaging directive. The 2030 scenario expanded the recycling yield by 3% relative to 2025 to meet the 55% recycling rate target. The recycling yields, which were the basis for the scenarios, are shown in Table 6.

Based on findings by Neubauer et al. (2020), the necessary resources for the recycling processes regarding consumption and input masses of recycling plants were calculated [16]. In addition, data from the literature were considered in the calculation (see Appendix F) regarding electricity, diesel and water consumption. The necessary resources for recycling multilayer films were assumed to be equal to those of existing recycling plants for film. This was done because reliable data for specialised recycling operations for multilayer polymer packaging films have been unobtainable because no method to fully deconstruct multilayer film into pure recyclable polymers is currently employed in recycling schemes [8,17,18]. Based on the input mass, the determined consumption of resources is listed in Table 7.

#### 2.4.8. Energy Recovery: Waste Incineration Plant

Any leftovers from sorting and recycling were used for energy recovery via transport processes. The “EU-28: Plastic packaging in municipal waste incineration plant” (GaBi) dataset was used to model the energy recovery process. The energy recovered from the incineration process was used to substitute the necessary primary energy supply, and the resulting steam was not used (steam sink).

The dataset used represents treatment in a waste-to-energy plant with dry flue gas scrubbing and Selective Catalytic Reduction as NOx removal techniques. The energy balance of the combustion model reflects the average situation in the European Union and takes the heat input of the specific waste into account. Emissions are calculated based on transmission coefficients and initial waste compositions are representative of European plant data. The dataset includes all relevant process steps for thermal treatment and corresponding processes, such as the disposal of waste air treatment residues or metal recycling. The inventory is essentially based on industrial data and is supplemented by secondary data where necessary. The system is partially closed (open outlets of electricity and steam). The electricity and steam flows need to be connected and adjusted to local conditions in order for these credits to be considered. Credits for recovered metals are already included.

#### 2.4.9. Energy Supply

Electrical Power: The power required in the individual process steps was mainly provided by primary energy sources (renewable/non-renewable). The composition of the energy sources in Austria and the environmental impact of electricity production was quantified by the “AT: Electricity grid mix” dataset and modelled as a process.

Natural Gas: The provision of the required amount of natural gas in the sorting process was represented by Thinkstep’s “Austria (AT): Natural gas mix” dataset and modelled as a process.

Diesel: The amount of fuel for transport and recycling processes was mapped by Thinkstep’s dataset “EU-28: Diesel mix at refinery” and modelled as a process.

#### 2.4.10. Transport

Transport processes were integrated into the accounting information to show the flow of goods in the processes. A transport model was created to map the recycling of a product in Austria and determine the average transport distances (cf. Appendix G). For this purpose, the shortest distances between selected locations of the individual processes were determined and averaged using Google Maps, considering the motorway connection for trucks. The utilisation rates of the transport vehicles were taken from the data from Öko-Institut e.V. et al. (2016), adopted for known transport routes. Unknown degrees of utilisation were estimated by assuming an optimised transport of products, considering the given literature values for similar transport processes [19]. The assignment of the transport vehicles or the data records to the transport routes was made at our discretion. The determined transport data are listed in Table 8.

### 2.5. Impact Assessment

The impact assessment was carried out using GaBi software, version 9.2.1.68 by Sphera Solutions GmbH (Leinfelden-Echerdingen, Germany). To calculate the selected environmental impacts in the target and scope of the investigation, the following assessment method and the following impact categories selected are detailed in the following.

### 2.6. Evaluation Method

Developed by the Centrum voor Mileukunde (CML), the CML 2001 method is an ecologically oriented information and decision-making tool for creating a life cycle assessment in accordance with DIN EN ISO 14040. CML 2001 quantifies the environmental impacts of the processes from the inventory analysis, links them to the selected impact categories and assigns them to the impact indicators, GWP and ADPF. The impact-oriented assessment method links the environmental impacts, considering the respective impact category over 100 years and includes the impact categories of climate change and resource consumption, which were chosen to compare the different scenarios.

#### 2.6.1. Impact Category and Impact Indicators

The impact categories considered included climate change, with the impact indicator GWP, and resource consumption, with the impact indicator fossil abiotic resource depletion.

#### 2.6.2. Impact Categories

In the following the impact categories considered in the course of the LCA are explained:


Global Warming Potential


The GWP effect parameter expresses the assessment of the intensification of the greenhouse effect. According to Frischknecht (2020), the GWP parameter considers the absorption coefficients for infrared thermal radiation, the residence time of the gases in the atmosphere and the expected emission development. The potential effects of 1 kg of greenhouse gas over 20 or 100 years compared to 1 kg of CO_2_ are determined and converted into equivalent CO_2_ emissions (kg CO_2_-eq) and shown in Table 9 [10].


Abiotic Resource Depletion (ADP)


The ADP for fossil resources is expressed in MJ as the quantity of resources consumed relative to the resources depleted [21].

## 3. Results

### 3.1. Occurrence and Treatment of Plastic Waste in Austria

The amount of plastics produced in Europe in 2018 was around 62 million tons (t), around 17% of global plastics production [1]. The packaging industry played the leading role in the demand for plastics with 39.9%. The PP, PET and polyethene (PE), in particular, were leaders in this segment [1].

In 2018, around 29.1 million t of post-consumer plastic waste was collected in Europe, resulting in an increase in waste by 19% compared to 2006 with 24.5 million tons. In Figure 10, it can be seen that the post-consumer plastic waste generated in Europe in 2018 was 32.5% recycled, 42.6% energetically recovered and 24.9% landfilled [1].

In the EU in 2018, 41.4% of the collected plastic packaging waste was recycled, 21.5% landfilled, and 37.0% energetically recovered, as shown in Figure 8. Only 0.1% was used for other purposes. The recovery processes were depicted according to the Waste Framework Directive [22].

In 2019, 1.5 million tonnes of plastic waste were exported from the EU, and most of this plastic waste was shipped to Asia. The export volume to China was around 1.4 million tonnes of plastic waste in 2016, which fell to 14,000 tonnes in 2019 due to an import ban on certain types of waste [23].

The volume of plastic packaging waste in the European Union (EU27) grew to around 14.8 million t per year between 2009 and 2018, corresponding to a per capita volume of 33.2 kg/PE, as shown in Figure 11.

In 2018, around 5 million tons of plastic recyclates (regenerates) were produced in Europe, and 80% of this flowed into European plastics production to create new products. With a share of 24% in recyclates, the packaging industry followed the construction industry with a share of 46% [24].

In 2018, Austria’s primary plastic waste weighed around 0.95 million tonnes. Plastic waste is subdivided into sorted plastics, solid waste-containing plastic and a remainder. This remainder, which accounts for around 2% of the total waste, consists of plastics in paints and varnishes, plasticisers, and plastic sludge. About 18% was single-variety plastics, such as plastic foils, polyolefin waste and plastic containers, and around 80% was solid waste-containing plastic, such as bulky waste, old tyres, municipal waste and similar commercial waste. As can be seen in Figure 12, 26% of primary plastic waste was recycled in 2018, 72% was used to generate energy, and 2% was landfilled as part of other types of waste [25].

In 2018, Austria’s volume of plastic packaging waste was approximately 34.2 kg per capita, totalling around 302,000 t. In 2018, 68% of the packaging waste was energetically recovered and 32% recycled, as apparent in Figure 12 [22].

According to van Eygen (2018), approximately 69,000 t of plastic foils were generated as waste in 2013. Around 52,000 t of plastic film waste were collected separately and sent to a sorting and processing process [5]. This result corresponds to a collection rate of around 75%. Roughly 17,000 t of pre-sorted plastic film waste were then sent to a recycling process, out of which roughly 12,000 t of regranulate were produced. Around 18% of all plastic film waste was mechanically recycled in 2013, and 82% was used in energy recovery [5].

### 3.2. Life Cycle Assessment

In Table 10, the impact assessment results using the CML 2001 assessment method are listed. The GWP and the ADPF of all evaluated scenarios are presented in Table 10. The deviations were given in the respective unit and the percentage deviation.

As shown in Table 10, Scenario 1 causes a GWP of 3027 kg CO_2_-eq and an ADPF of 50,978 MJ.

The change in collection results in a reduction of 210 kg CO_2_-eq from SQ, which equals roughly 6%. Simultaneously the ADPF was reduced by 3790 MJ, or 7%.

The graphical representations of the results, including the individual process effects, are shown in Figure 13. In addition to the overall impact, six individual process impacts were plotted. The individual process effects were chosen following their most considerable contribution to the overall effect of the SQ scenario. The group “rest” of the GWP, in Figure 13, includes the total effects of the individual processes, such as transport processes and primary energy sources, which are not shown due to their low GWP. The group “rest” of the ADPF, in Figure 14, includes the total effects of the individual processes not shown due to their low ADPF, such as transport processes and individual waste incineration processes.

Note the reduction in GWP in the SQ scenario and Scenario 2 due to the “rest” group processes. This reduction mainly includes the effects of the substitute energy supply through the combustion processes concerning the primary energy supply.

The production of PE and the incineration of mixed solid waste (MSWI) and packaging films are the most significant contributors to the GWP. These are substantially reduced by implementing a recycling scheme for polymer films, as shown in Figure 13. The production of virgin PE and PP in Scenario 3 yields no emission because recycled polymers substitute virgin materials in this ideal scenario. The increased demands for fuels by sorting and transportation of the increased recycling materials yields a minuscule amount of GWP and is not shown as a unique bar in the Figure 13.

#### 3.2.1. Global Warming Potential

Figure 15 shows the results of the GWP and the deviations from the SQ scenario and the 2025 scenario. For Scenario 2, with a GWP of 3027 kg CO_2_-eq, a reduction of 6% compared to the SQ can be seen.

Comparing the GWP of the scenarios shows that the SQ of plastics recycling causes the most significant environmental impact. The product lifecycle of 1000 kg of plastic films, consisting of mono- and multilayer films, causes a GWP of 3237 kg CO_2_-eq in scenario SQ. The driving processes are the PE production and the waste incineration of the monolayer films, followed by the incineration of the municipal waste.

The GWP in Scenario 2 was decreased by 6% relative to the SQ. This reduction in GWP is due to an improved collection rate and amounted to an absolute reduction of 3027 kg CO_2_-eq when compared to the GWP of the SQ.

The increased mass accumulation of waste to be sorted and recycled is reflected in a reduction in the impact of PE production and the increase in the contribution of monolayer film waste incineration to the GWP. The incineration of municipal waste is removed from the balance due to the separate collection of the plastic foils.

Scenario 3 shows that recycling the monolayer films reduces the GWP by 63%, from the GWP of the SQ to 1191 kg CO_2_-eq This reduction is due to the exclusion of waste incineration of monolayer films and the reduced production volume of PE due to the increased amount of regranulate. The waste incineration process of the multilayer films is the primary source of emissions of greenhouse gases.

Scenario 4 has the lowest GWP and is the ecologically best scenario variant. Due to the simulated recycling of all plastic films, the GWP is 90% lower than the SQ, at 335 kg CO_2_-eq per 1000 kg produced plastic foils. This reduction is achieved by substituting regranulates for all primary virgin materials used in packaging production, generating a closed material cycle. This substitution eliminates the need to produce virgin materials and, thus, reduces the production-related effects to zero. The optimisation of the collection rate, sorting depth and recycling yield eliminates the need for incineration. In Scenario 4, primary energy becomes the predominant emission source for the GWP.

The optimisations considered in Scenario 2025, the collection rate, sorting depth and recycling yield, lead to a reduction of the greenhouse potential by 34%, to the SQ, to 2124 kg CO_2_-eq, as can be seen in Figure 15.

The 2030 scenario is an extension of the 2025 scenario. As a result, a prescribed recycling rate of 55% by 2030 reduces the GWP by 40% to 1944 kg CO_2_-eq compared to the SQ. The prescribed increase in the recycling rate to 55% leads to a reduction of the GWP by 8%. This reduction is caused by the further reduction of process emissions by 2030.

#### 3.2.2. Abiotic Resource Depletion Fossil

The SQ scenario maps the maximum ADPF with a value of 54,769 MJ. This value is taken as the benchmark all other scenarios will be compared against. The predominant consumption of fossil resources in the SQ occurs through the production process of PE. Compared to the SQ, Scenario 2 shows an ADPF of 50,978 MJ, which results in a 7% reduction relative to the SQ. This reduction of ADPF in Scenario 2 is due to the increased collection rate. This increased collection rate allows for increased substitution of virgin primary plastic granules with regranulates, reducing resource consumption for the production of virgin polymers.

The improved recycling of the monolayer foils considered in Scenario 3 reduces the ADPF by 66%, to the SQ, to 18,515 MJ. This recycling and the resulting increase in PE regranulate leads to a noteworthy reduction in the consumption of resources in virgin PE production. Consequently, the resource consumption of Scenario 3 is dominated by the necessary consumption for the manufacture of multilayer films. The production of multilayer films replaces the production of monolayer films as the predominant source of resource consumption because the produced regranulates reduce the necessary production volume for monolayer films.

Scenario 3 shows that the ADPF can be reduced by 93% to 3703 MJ by recycling the mono- and multilayer films. As a result of substituting virgin plastic granules from PE and PP production with regranulates from the recycling process, this consumption of resources is reduced to zero. The consumption of 3703 MJ of resources is based almost entirely on the provision of energy.

Scenario 2025 shows that optimising the waste processing procedures and the collection operation to a point where the mandated recycling quota is met leads to a reduction of the ADPF of 36% to the SQ.

Meeting the required 50% recycling rate for plastic packaging waste by 2025 in the recycling of films results in the consumption of fossil resources of 35,185 MJ. In the 2025 scenario, the PE production process causes substantial resource consumption, followed by the PP production process.

As a result of adhering to the stipulated recycling rate of 55% by 2030, the 2030 scenario shows reduced ADPF to SQ by 42%, resulting in a consumption of 32,028 MJ. Increasing the recycling rate to 55% leads to a reduction of the ADPF by around 9% from 2025 to 2030. This reduction is caused by the further reduction in the process consumption of fossil granulate production by 2030 compared to the 2025 scenario.

Figure 16 shows the abiotic resource depletion of the individual scenarios and the deviations from the SQ scenario.

## 4. Conclusions

An LCA allows for the investigation and quantification of environmental changes under changed process parameters. The LCA presented in this work determined the resulting environmental impacts of monolayer and multilayer films during their lifecycle on GWP and ADPF. For this purpose, based on the SQ, scenarios with changed parameters regarding the collection rate, sorting depth and recycling yield were created and evaluated. Finally, scenarios for evaluating the necessary improvements in the waste recycling processes to achieve the statutory recycling rate targets were examined, and the environmental impacts were considered. A “functional unit” of 1000 kg of plastic film waste, generated as post-consumer waste in Austria and recorded in the light packaging collection’s collection and recycling system, was selected.

The results of the LCA showed the general trend toward reducing environmental impacts by optimising collection, sorting and recycling. The GWP of the SQ was 3237 kg CO_2_-eq. This SQ was used as the basis for comparing the other scenarios. Furthermore, the ADPF in the SQ was determined to be 54,769 MJ. Increasing the collection rate for the separate collection of plastic foils in Scenario 2 reduced the GWP by 6% to 3027 kg CO_2_-eq and the ADPF by 7% to 50,978 MJ. Substantial improvements were achieved by recycling monolayer films made possible by the ejection of multilayer films from the material recycling process. Considering this, recycling in Scenario 3 led to a reduction of the GWP by 63% or 2,046 kg CO_2_-eq to 1191 kg CO_2_-eq and the ADPF by 66% or 36,253 MJ to 18,515 MJ. The ecologically best result was achieved by the closed cycle management of monolayer and multilayer films. Based on the SQ, the GWP could be reduced by 90% or 2902 kg CO_2_-eq to 335 kg CO_2_-eq. The ADPF fell by 93% or 51,066 MJ to 3703 MJ. The transition from the SQ to the circular economy caused a shift in the emission-relevant processes of the greenhouse potential from production or combustion to energy supply. Given the politically stipulated recycling rate for packaging plastics of 50% by 2025 and 55% by 2030, optimisations should be sought in all areas of waste management. The sorting depth was identified as the most influential parameter. Increasing collection rate and recycling yield by 20% demands a surge in sorting depth from 34% to 63.4%. These optimisations in the 2025 scenario reduced the GWP by 1113 kg CO_2_-eq or 34% to 2124 kg CO_2_-eq and the ADPF by 19,584 MJ or 36% to 35,185 MJ.

A recycling rate of 55% by 2030 requires improving the collection rate by 20% and the recycling yield by around 23% based on the SQ. This increase means doubling the sorting depth from 34% to around 68%.

Comparing the 2030 scenario to the SQ scenario showed that these improvements reduced the GWP or ADP by around 40% to 1944 kg of CO_2_-eq or 42% to 32,028 MJ. The reductions in environmental impact caused by increasing the recycling rate to 55% from 2025 to 2030 could be achieved with 180 kg of CO_2_-eq or 3157 MJ can be determined.

The LCA showed that improvements in improved sorting and increased recycling of mono- and multilayer films are desirable and necessary. In addition to the political and social efforts to transform waste management into a circular economy, the strive to provide renewable energy sources should be intensified. This approach would conserve primary resources and facilitate the transition to a circular economy.

## Figures and Tables

**Figure 1 polymers-14-03620-f001:**
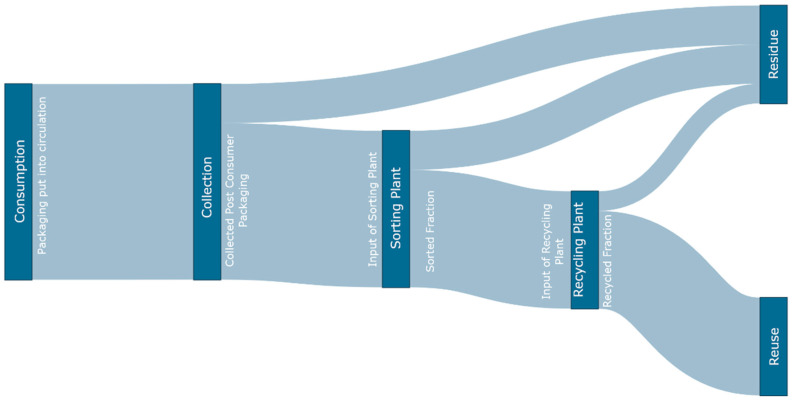
Sankey diagram laying out the different inputs and outputs of the recycling system.

**Figure 2 polymers-14-03620-f002:**
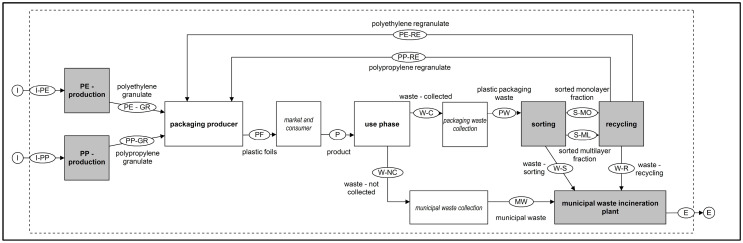
Product lifecycle of plastic films.

**Figure 3 polymers-14-03620-f003:**
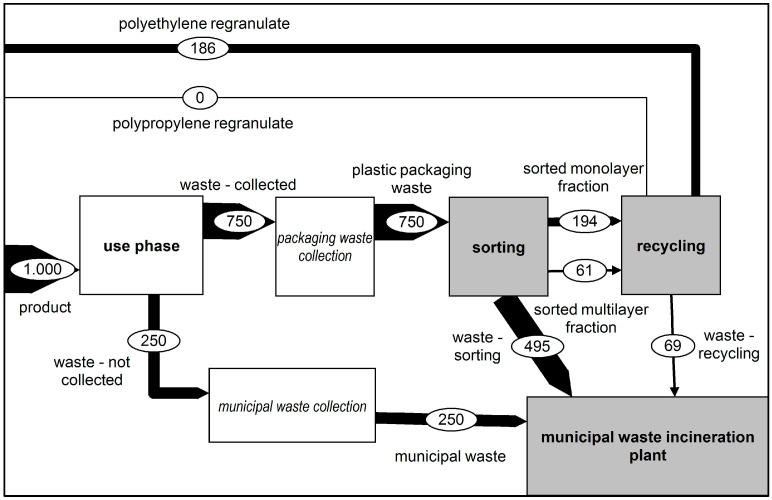
Representation of material flows—Status Quo Scenario. Composition of the functional unit and the utilisation from van Eygen et al., 2018 [5].

**Figure 4 polymers-14-03620-f004:**
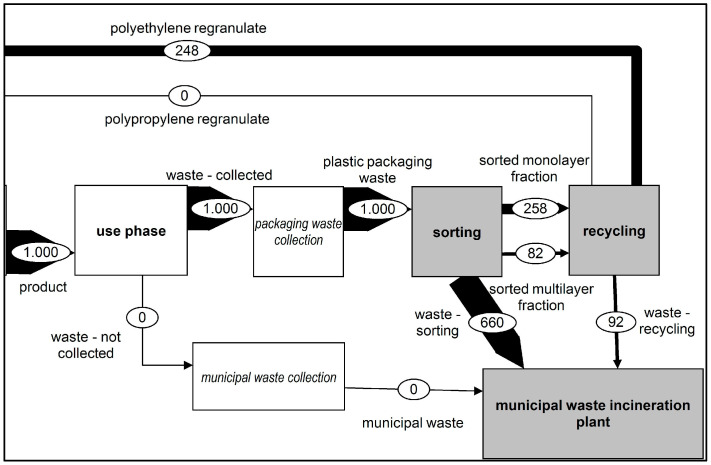
Representation of the changed material flows—Scenario 2.

**Figure 5 polymers-14-03620-f005:**
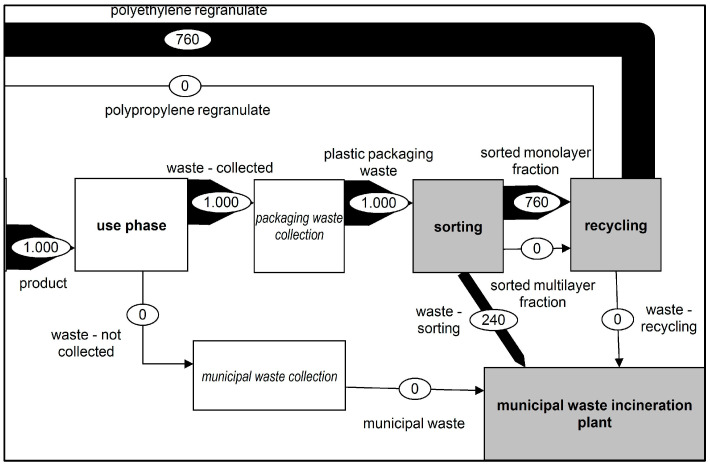
Representation of the changed material flows—Scenario 3.

**Figure 6 polymers-14-03620-f006:**
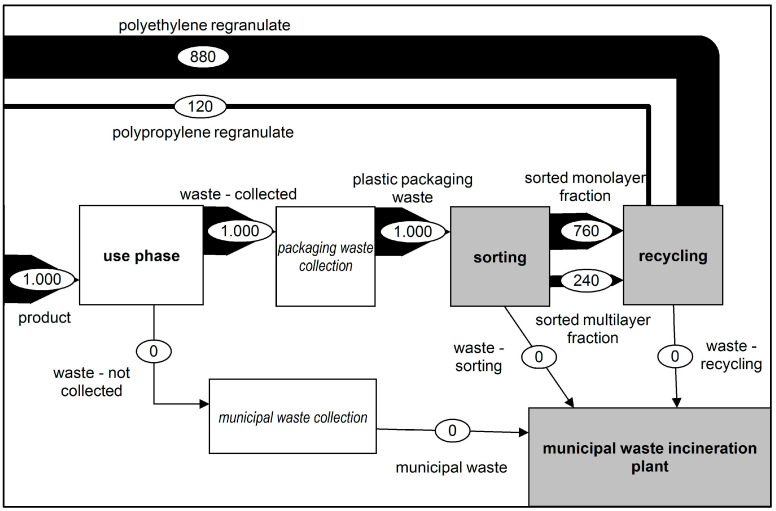
Representation of the optimised material flows—Scenario 4.

**Figure 7 polymers-14-03620-f007:**
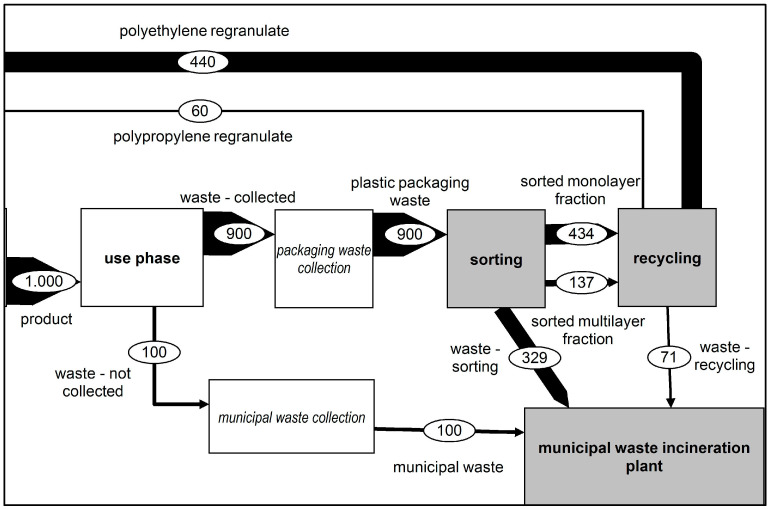
Representation of the material flows—2025 scenario.

**Figure 8 polymers-14-03620-f008:**
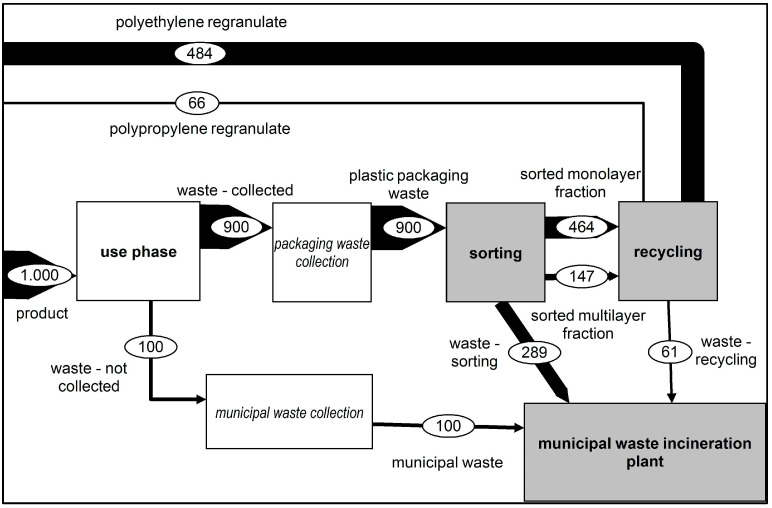
Representation of the material flows—2030 scenario.

**Figure 9 polymers-14-03620-f009:**
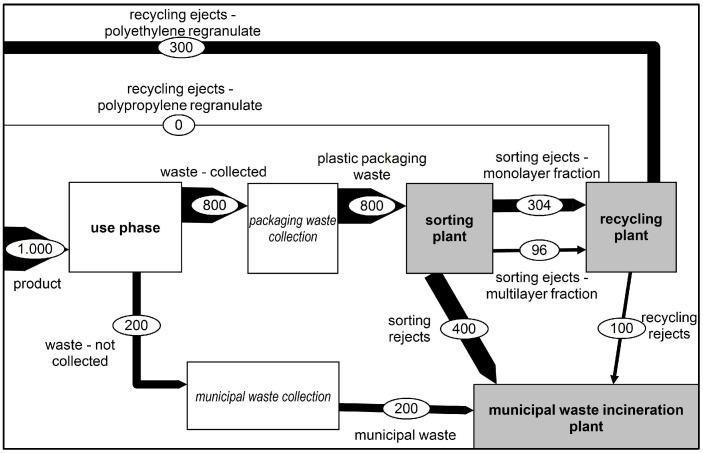
Representation of the material flows in Scenario 7.

**Figure 10 polymers-14-03620-f010:**
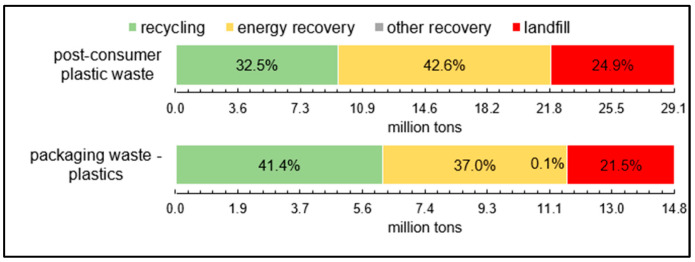
Treatment rates of post-consumer plastic waste and plastic packaging waste in the European Union 2018 [22].

**Figure 11 polymers-14-03620-f011:**
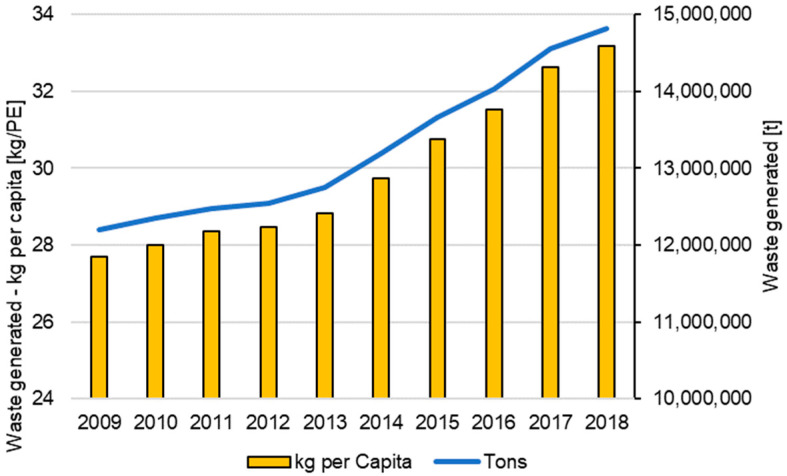
Plastic packaging—average waste generated in the EU from 2009 to 2018 [22].

**Figure 12 polymers-14-03620-f012:**
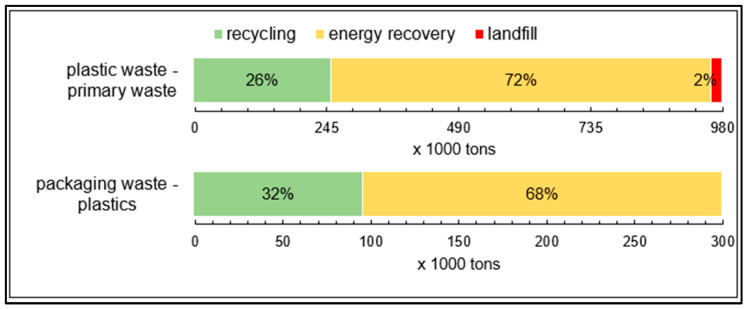
Treatment rates of primary plastic waste and plastic packaging waste in Austria in 2018, own illustration [22,25].

**Figure 13 polymers-14-03620-f013:**
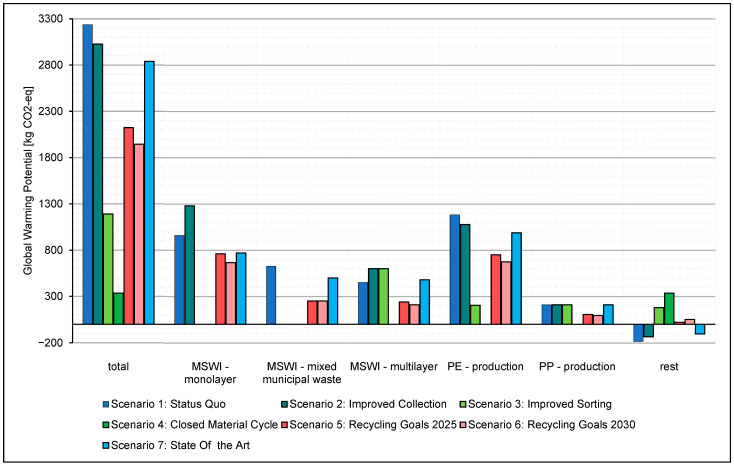
GWP of the individual processes over the product lifecycle. Mixed Solid Waste Incineration.

**Figure 14 polymers-14-03620-f014:**
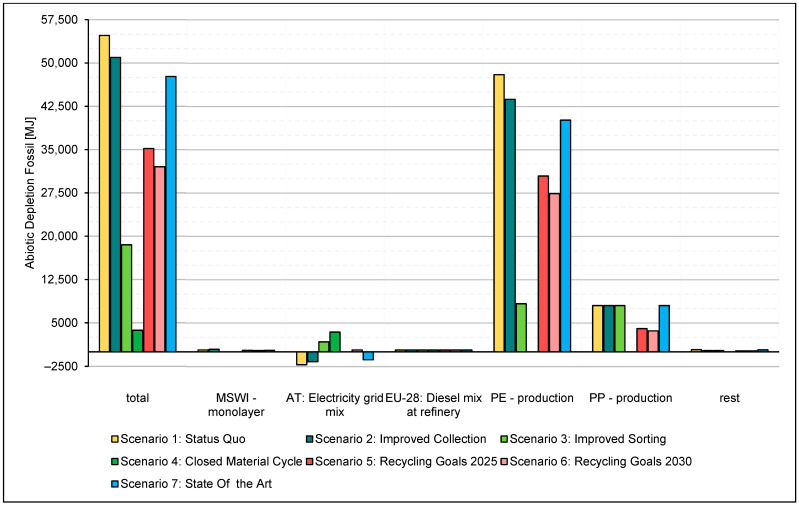
ADPF of the individual processes over the product lifecycle.

**Figure 15 polymers-14-03620-f015:**
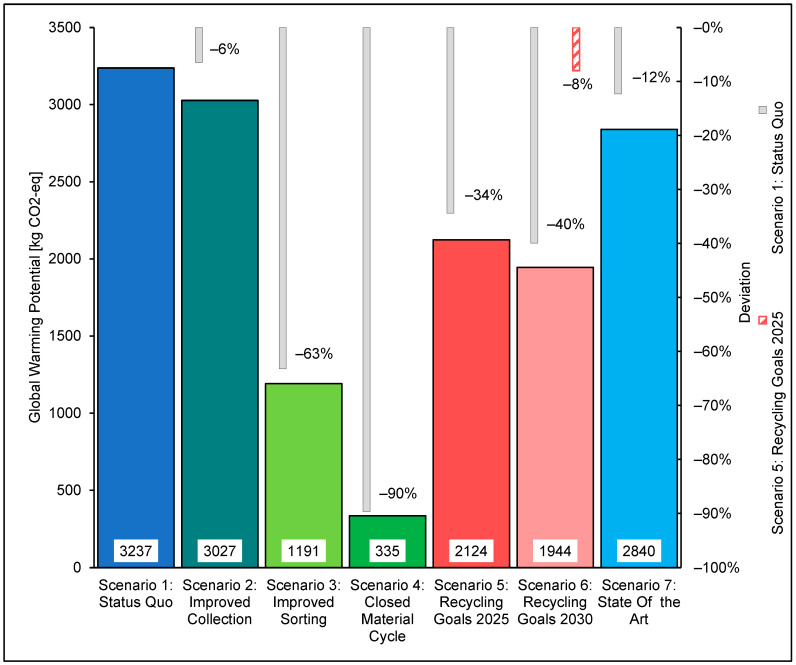
Global warming potential (GWP)–scenario overview.

**Figure 16 polymers-14-03620-f016:**
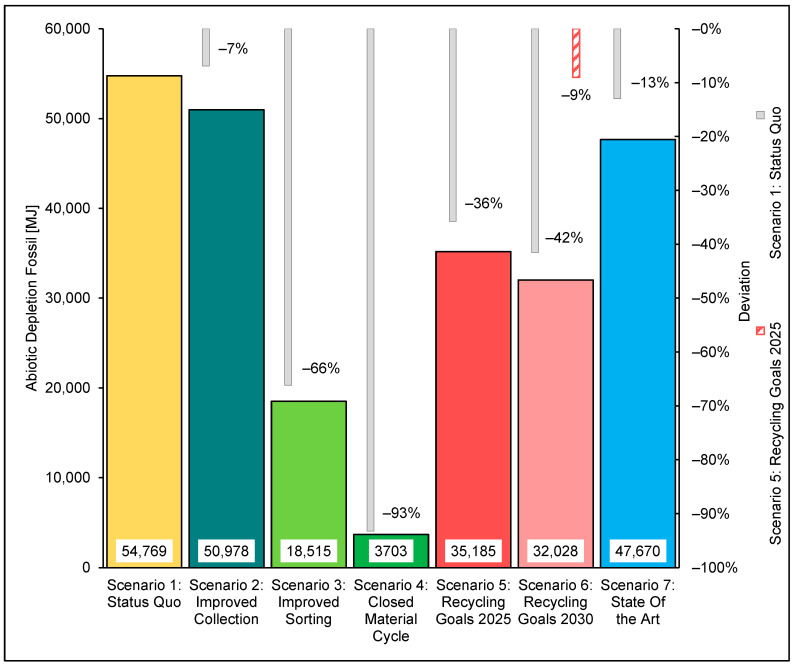
Abiotic Resource Depletion ADPF—Scenarios Overview.

**Table 1 polymers-14-03620-t001:** Overview of the collection rates, sorting depths and recycling yield in the evaluated Scenarios.

Scenario	Collection Rate	Sorting Depth		Recycling Yield	
		Monolayer Films	Multilayer Films	Monolayer Films	Multilayer Films
1 SQ	75%	34%	34%	73%	0%
2 IC	100%	34%	34%	73%	0%
3 IS	100%	100%	0%	100%	0%
4 CMC	100%	100%	100%	100%	100%
5 2025	90%	63%	63%	88%	88%
6 2030	90%	68%	68%	90%	90%
7 SOA	80%	50%	50%	75%	0%

**Table 2 polymers-14-03620-t002:** Material composition of the plastic films under consideration and their share in the material flow [4].

	Material Composition		Share in the Material Flow
Monolayer Film	LDPE 100% by weight	/	76% by weight
Multilayer Film	LDPE 50% by weight	PP 50% by weight	24% by weight

**Table 3 polymers-14-03620-t003:** Collection rates of the LVP collection.

	Collection Rate [t_waste collected separately_/t_waste disposed of_]						
	Scenario 1 SQ	Scenario 2 IC	Scenario 3 IS	Scenario 4 CMC	Scenario 5 2025	Scenario 6 2030	Scenario 7 SOA
Plastic Films	75%	100%	100%	100%	90%	90%	80%

**Table 4 polymers-14-03620-t004:** Modelled sorting depths of the target fractions for every scenario.

	[t_output target fraction_/t_input_]						
	Scenario 1 SQ	Scenario 2 IC	Scenario 3 IS	Scenario 4 CMC	Scenario 5 2025	Scenario 6 2030	Scenario 7 SOA
Monolayer Films	34%	34%	100%	100%	63.4%	67.9%	50%
Multilayer Films	34%	34%	0%	100%	63.4%	67.9%	50%

Annotation. The SQ’s sorting depth calculation can be seen in Appendix D.

**Table 5 polymers-14-03620-t005:** Resource consumption of the sorting.

Operating Resources	Consumption	Unit
Electricity	63.97	kWh/t_INPUT_
Gas	1.49	kWh/t_INPUT_

Annotation. The calculation of the resource consumption for the sorting can be seen in Appendix D. Note on the resource calculation of the “Saubermacher Dienstleistungs AG” sorting system in Appendix D.

**Table 6 polymers-14-03620-t006:** Recycling yield of the target fraction depending on the scenarios.

	Recycling Yield [t_REGRANULATE_/t_INPUT_]						
	Scenario 1 SQ	Scenario 2 IC	Scenario 3 IS	Scenario 4 CMC	Scenario 5 2025	Scenario 6 2030	Scenario 7 SOA
Monolayer Films	73% (96%)	73% (96%)	100%	100%	87.6%	90%	75%
Multilayer Films	0%	0%	0%	100%	87.6%	90%	0%

Annotation. The SQ’s recycling yield calculation can be found in Appendix F.

**Table 7 polymers-14-03620-t007:** Resource consumption of the recycling process.

Operating Resources	Consumption	Unit
Electricity	629.83	kWh/t_INPUT_
Diesel	12.41	kWh/t_INPUT_
Water	2.25	m^3^/t_INPUT_

Annotation. The calculation of the resource consumption for recycling can be seen in Appendix F.

**Table 8 polymers-14-03620-t008:** Specific transport distances, utilisation rates and vehicles.

Fraction	Route	Vehicle	Workload	Distance
			[%]	[km]
PE-GR/PP-GR	Basic material manufacturer → packaging producer	X ^a^	90	293
W-C	Use phase → packaging waste collection	0 ^b^	50	10
PW	Packaging waste collection → sorting	X	75	100
S-MO/S-ML	Sorting → recycling	X	83	156
W-S	Sorting → municipal waste incineration plant	X	90	87
W-R	Recycling → municipal waste incineration plant	X	75	95
W-NC	Use phase → municipal waste collection	0	50	10
MW	Municipal waste collection → municipal waste incineration plant	X	75	100
PE-RE/PP-RE	Recycling → packaging producer	X	90	104

^a^ GLO: Truck trailer, Euro 0–6 mix, 34–40 t gross weight/27 t payload capacity (GaBi). ^b^ GLO: Truck, Euro 5, 14–20 t gross weight/11.4 t payload capacity (GaBi). Annotation. The determination of the transport model data can be seen in Appendix G.

**Table 9 polymers-14-03620-t009:** GWP and temperature change potential (GTP) emission parameters [20] result from 1 kg CO2.

		GWP		GTP	
	Lifetime	Cumulative Forcing over 20 Years	Cumulative Forcing over 100 Years	Temperature Change after 20 Years	Temperature Change after 100 Years
CO_2_		1	1	1	1
CH_4_	12.4	84	28	67	4
N_2_O	121	264	265	277	234
CF_4_	50,000	4880	6630	5270	8040
HFC-152a	1.5	506	138	174	19

**Table 10 polymers-14-03620-t010:** Results and deviations of the impact assessment compared to the SQ.

Scenario	GWP			ADPF		
	Result	Deviation to SQ		Result	Deviation to SQ	
	[kg CO_2_-eq]	[kg CO_2_-eq]	[%]	[MJ]	[MJ]	[%]
Scenario 1 SQ	3237			54,769		
Scenario 2 IC	3027	−210	−6	50,978	−3790	−7
Scenario 3 IS	1191	−2046	−63	18,515	−36,253	−66
Scenario 4 CMC	335	−2902	−90	3703	−51,066	−93
Scenario 5 2025	2124	−1113	−34	35,185	−19,584	−36
Scenario 6 2030	1944	−1293	−40	32,028	−22,741	−42
Scenario 7 SOA	2840	−397	−34	35,185	−19	

## Data Availability

The data presented in this study are available on request from the corresponding author.

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
