# Peer review of "Lifecycle Assessment for Recycling Processes of Monolayer and Multilayer Films: A Comparison"

_polymers, 2022, doi:10.3390/polym14173620_

Round 1
Reviewer 1 Report
Comment on paper
This work presents an LCA of monolayer and multilayer films, with the aim of quantifying the environmental impacts derived from the change in the management of plastic film waste.
The text has many errors, which makes it difficult to read (sections, figures, tables, wrong references, etc.). The best scenario is the closed loop scenario, which is not viable, and the second-best scenario is the one that includes improvements in the classification, which has not been described correctly.
The results obtained are evident and expected, and the work does not provide a clear contribution.
In the conditions in which the work is currently, it does not have the expected quality to be published.
Specific Comments:
- Include the year in the references. The format of the references should be respected.
- Acronyms must be defined the first time they are used: ADPF.
- Correct the numbering of the formulas: Formula 2.
- What is Article 11a? Its definition is missing.
- The calculations and definitions section is not clear. The variety of terms could confuse the reader. To avoid this, it would need to be rewritten, or perhaps accompanied with a figure laying out the different inputs and outputs of the recycling system (packaging put into circulation, collected post-consumer packaging, input of sorting plant, input of recycling plant, etc.).
- The appendices should be in order of mention.
- Reference to GaBi. The explanation of the software is subsequent to its first mention
- It would be appropriate to highlight the differences with respect to the base case to see more easily the improvements of the scenarios with respect to the base case.
- In Error! Reference source not found. Pg 9., Pg 14, Pg 18, etc.
- The acronyms used for the scenarios should be defined (SQ, IC, etc.).
- The numbering of the tables is not correct. There are two 1-tables and two 2-tables. The numbering of the sections is not correct (2.2, etc.). These errors make reading difficult.
- The appendices are pasted without any explanation.
- In Figure…? Pag 16
- The text does not clearly explain the costs and needs of new processes, such as thermal recovery.
Reviewer 2 Report
The manuscript is interesting, well structured and well written. Even if many LCA have been conducted on recycling of plastics, this manuscript presents novelty and the study has a good scientific height. It can be published after some minor editions.
· The authors use the wording “lifecycle assessment” and “lifecycle analysis”. I will recommend to use just “life cycle assessment” because it is the nomenclature used in the ISO standard 14040.
· First paragraph on page 2: “….300,000 t of waste generated…”. You should describe what kind of waste you mean.
· On page 2 (end of Chapter 1) “ADPF” is first mentioned in the second last paragraph, but is explained in the next paragraph (last paragraph of chapter 1). All notations and acronyms should be explained directly at the first time they are used.
· Equation 1: The concept “Packaging put into circulation” needs a better explanation. “put into circulation” may be misinterpreted as “put to recycling”
· Links in the manuscript is not updated. “Error! Reference source not found” occurs on several places in the manuscript
· Page 12. “2D recycling” should be explained or changed into “recycling of plastic films” or similar.
· Table 4. The concept “sorting depth” needs an explanation.
Reviewer 3 Report
The authors performed a life cycle assessment and determined the resulting environmental impacts of monolayer and multilayer films during their life cycle on global warming potential and the abiotic resource depletion fossil. I recommend this paper be published in the Polymers Journal. To improve the manuscript, please consider the following comments:
1- For more clarification, add type of plastic films in the Introduction section.
2- In the Introduction section, bring out clearly the gap left that you want to fill in this study.
3- Add sources for formulas 1-4, and remove (%) at the end of these formulas, correct the number of formula 2
